# OpenReview forum: "Evaluating and Improving Cultural Awareness of Reward Models for LLM Alignment"
_ICLR.cc/2026/Conference — ICLR 2026 Poster_

### Official Review · Reviewer_9wq4 · 2025-10-27

**Soundness:** 3
**Presentation:** 3
**Contribution:** 3
**Rating:** 6
**Confidence:** 3

**Summary:**

In this paper, the authors focused on the important problem of cultural awareness in reward models (RM). Their contributions include proposing a multilingual, culturally aware reward model (CARB), covering 10 typologically diverse cultures and 4 key cultural dimensions. By analyzing over 20 classifier-based and generative RMs, the authors demonstrate that generative RMs outperform classifier-based ones in capturing cultural nuance. Their results further highlight that value alignment as the most difficult domain. The paper also contributes to improving the RMs with Think-as-Locals, a dual reward function to improve the training of culturally aware RMs.

**Strengths:**

S1: The paper focused on a timely and the important problem of cultural awareness in reward models.

S2: The proposed CARB benchmark explicitly designed for cultural awareness in reward models, filling a genuine research gap, unlike prior multilingual benchmarks ignoring this aspect. CARB covers 10 typologically diverse cultures and 4 key cultural dimensions.

S3: The paper provides comprehensive evaluations on both classifier-based and generative RMs (over 20 models), yielding valuable insights on comparative performance. Nice causal and spurious correlations analysis provide interesting insights of why current RM failed at cultural awareness.

S4: The proposed dual reward function (correctness + criteria appropriateness) is justified and experimentally validated to show effectiveness.

**Weaknesses:**

W1: Although some human validation exists (L173), the dataset’s independence from model-generated bias and with human cultural experts in judging CARB as well as RM judgments.

W2: While carefully filtered and validated by some humans, the CARB benchmark heavily depends on LLM generated data. Including authentic human generated examples would strengthen the benchmark.

W3: [Minor] Despite the coverage of 10 cultures, CARB favours high(er)-resource languages. Low-resource cultures (e.g., African or Indigenous) remain underrepresented, including some of the lower-resource cultures could further strengthen the work and show generalization.

**Questions:**

Comment: Values in Fig. 3 are a bit difficult to see, consider using a darker colour for the grid.

---

> ### Author Response · Authors · 2025-11-21
> **Response to Reviewer 9wq4 (1/3)**
>
> We sincerely thank you for providing thoughtful and constructive feedback. Please kindly find our response to your comments below. We have revised the manuscript by adding the new evaluation experiments on native-speaker–written best-of-n sets in **Section 4.2** and **Appendix C.7** and refining **Figure 3** for improved clarity. All revisions made to the paper are highlighted in blue for your ease of reference. We hope that our response satisfactorily addresses the issues you raised. Please feel free to let us know if you have any additional concerns or questions :-)
>
> ---
>
> > W-1 Although some human validation exists (L173), the dataset’s independence from model-generated bias and with human cultural experts in judging CARB as well as RM judgments.
> >
>
> We appreciate the reviewer’s comment; however, the meaning of the sentence in **W-1** remains somewhat unclear to us. We interpret it as a concern regarding **the reliability of our dataset, particularly its independence from model-induced bias**. Because this issue closely relates to the points raised in **W-2**, we have integrated our detailed response—and additional evaluations using authentic human-written examples—into that section. To directly address the concern about bias, we conducted further experiments based on **authentic human-written examples**. The results are highly consistent with our original CARB evaluation (Section 4.2), demonstrating that potential model-generated bias **does not undermine the validity or conclusions** of our study.
>
> **If this interpretation does not fully capture your concern, please kindly let us know, and we will provide a dedicated response immediately.**
>
> ---

---

> ### Author Response · Authors · 2025-11-21
> **Response to Reviewer 9wq4 (2/3)**
>
> > W-2 While carefully filtered and validated by some humans, the CARB benchmark heavily depends on LLM generated data. Including authentic human generated examples would strengthen the benchmark.
> >
>
> Thank you for your constructive feedback regarding our data generation and validation process.
>
> We would first like to clarify a point from our original manuscript. As described in **Section 3** and **Appendix B.3**, our benchmark refinement and validation process already incorporated substantial human oversight. Specifically, three human annotators manually refined the prompts and assessed response appropriateness under the guidance of two professional translation experts from Lan-bridge—an ISO-recognized institution providing qualified translation services—who served as instructors and evaluators.
>
> To directly and empirically address your concern about LLM-generated data (W-1, W-2), we conducted a **new evaluation study** during the rebuttal period using 100% human-authored data. **We recruited 10 new native-speaker evaluators—international students currently studying at Xi’an International Studies University**, each representing one cultural group and holding at least an undergraduate degree. We tasked them with **refining original prompts and** **writing completely new, authentic "chosen" and "rejected" completions** for a randomly sampled subset of 100 prompts per language. This created a new, parallel, 100% human-authored test set.
>
> We then re-evaluated the RMs on this new, human-written benchmark. As shown in the table below, the results confirm the validity of CARB:
>
> - **Increased Difficulty:** As expected, authentic human responses contain nuances that are harder for RMs to distinguish, resulting in lower absolute accuracy scores across the board (e.g., a ~12% drop for the top model).
> - **Preserved Rankings (Key Finding):** Crucially, the **relative performance trends remain consistent**. The model rankings on the human set align with those on the original CARB dataset. The models' relative strengths and weaknesses on specific languages (e.g., stronger performance on Chinese/German, weaker on Vietnamese/Japanese) also remain consistent.
>
> - Results on the Human-Authored Validation Set.
>
>
>     | **Model** | **Arabic** | **Chinese** | **English** | **German** | **Japanese** | **Korean** | **Russian** | **Spanish** | **Thai** | **Vietnamese** | **Overall** |
>     | --- | --- | --- | --- | --- | --- | --- | --- | --- | --- | --- | --- |
>     | gpt_4_1_2025_04_14 | 0.63 | 0.61 | 0.58 | 0.67 | 0.59 | 0.58 | 0.59 | 0.58 | 0.59 | 0.53 | 0.595 |
>     | Qwen3-235B-A22B-Instruct-2507 | 0.58 | 0.63 | 0.59 | 0.6 | 0.58 | 0.56 | 0.61 | 0.6 | 0.64 | 0.55 | 0.594 |
>     | deepseek-reasoner | 0.56 | 0.67 | 0.54 | 0.68 | 0.62 | 0.59 | 0.57 | 0.55 | 0.59 | 0.49 | 0.586 |
>     | deepseek-chat | 0.56 | 0.66 | 0.57 | 0.61 | 0.58 | 0.54 | 0.63 | 0.53 | 0.58 | 0.58 | 0.584 |
>     | Qwen2.5-72B-Instruct | 0.5 | 0.6 | 0.55 | 0.63 | 0.51 | 0.52 | 0.55 | 0.53 | 0.61 | 0.57 | 0.557 |
>     | Skywork-Reward-Gemma-2-27B | 0.57 | 0.58 | 0.58 | 0.58 | 0.53 | 0.51 | 0.55 | 0.51 | 0.63 | 0.48 | 0.552 |
>     | Qwen2.5-32B-Instruct | 0.55 | 0.64 | 0.54 | 0.62 | 0.5 | 0.5 | 0.57 | 0.51 | 0.59 | 0.46 | 0.548 |
>     | gpt_4o_2024_08_06 | 0.54 | 0.54 | 0.61 | 0.63 | 0.5 | 0.53 | 0.5 | 0.54 | 0.55 | 0.48 | 0.542 |
>     | Skywork-Reward-Gemma-2-27B-v0.2 | 0.52 | 0.57 | 0.59 | 0.6 | 0.51 | 0.54 | 0.54 | 0.51 | 0.57 | 0.47 | 0.542 |
>     | INF-ORM-Llama3.1-70B | 0.49 | 0.61 | 0.61 | 0.54 | 0.52 | 0.55 | 0.48 | 0.49 | 0.53 | 0.38 | 0.52 |
>
> - Results on the Original CARB Subset (LLM-Generated + Human Refined)
>
>
>     | **Model** | **Arabic** | **Chinese** | **English** | **German** | **Japanese** | **Korean** | **Russian** | **Spanish** | **Thai** | **Vietnamese** | **Overall** |
>     | --- | --- | --- | --- | --- | --- | --- | --- | --- | --- | --- | --- |
>     | Qwen3-235B-A22B-Instruct-2507 | 0.75 | 0.8 | 0.76 | 0.77 | 0.75 | 0.73 | 0.78 | 0.77 | 0.81 | 0.73 | 0.765 |
>     | gpt_4_1_2025_04_14 | 0.79 | 0.77 | 0.74 | 0.84 | 0.74 | 0.758 | 0.75 | 0.74 | 0.76 | 0.71 | 0.760 |
>     | deepseek-reasoner | 0.73 | 0.83 | 0.71 | 0.85 | 0.790 | 0.75 | 0.74 | 0.73 | 0.76 | 0.67 | 0.756 |
>     | deepseek-chat | 0.74 | 0.82 | 0.74 | 0.78 | 0.740 | 0.71 | 0.81 | 0.70 | 0.75 | 0.75 | 0.754 |
>     | Qwen2.5-72B-Instruct | 0.66 | 0.76 | 0.72 | 0.8 | 0.67 | 0.69 | 0.72 | 0.7 | 0.78 | 0.74 | 0.724 |
>     | Qwen2.5-32B-Instruct | 0.73 | 0.81 | 0.71 | 0.8 | 0.67 | 0.67 | 0.74 | 0.68 | 0.76 | 0.63 | 0.720 |
>     | Skywork-Reward-Gemma-2-27B | 0.73 | 0.74 | 0.75 | 0.74 | 0.7 | 0.68 | 0.72 | 0.68 | 0.8 | 0.64 | 0.718 |
>     | Skywork-Reward-Gemma-2-27B-v0.2 | 0.69 | 0.74 | 0.75 | 0.77 | 0.68 | 0.71 | 0.715 | 0.68 | 0.73 | 0.64 | 0.711 |
>     | gpt_4o_2024_08_06 | 0.71 | 0.71 | 0.77 | 0.80 | 0.66 | 0.70 | 0.67 | 0.71 | 0.71 | 0.65 | 0.709 |
>     | INF-ORM-Llama3.1-70B | 0.65 | 0.77 | 0.78 | 0.7 | 0.69 | 0.71 | 0.64 | 0.66 | 0.69 | 0.54 | 0.683 |

---

> ### Author Response · Authors · 2025-11-21
> **Response to Reviewer 9wq4 (3/3)**
>
> > Continuing response to W-2
> >
>
> This experiment provides strong evidence that **the conclusions in our paper remain robust when evaluated using exclusively human-authored data**. These empirical results further confirm that the original procedure **did not introduce model-generated biases** that might compromise the study’s validity. Overall, the close correspondence between results offers empirical support that **CARB is a reliable and high-fidelity proxy for genuine human cultural preferences**.
>
> **Action taken**: We have revised the manuscript to incorporate the full details and results of this human-authored validation study in **Section 4.2 and Appendix C.7**.
>
> ---
>
> > W-3 [Minor] Despite the coverage of 10 cultures, CARB favours high(er)-resource languages. Low-resource cultures (e.g., African or Indigenous) remain underrepresented, including some of the lower-resource cultures could further strengthen the work and show generalization.
> >
>
> We thank the reviewer for this constructive suggestion. As a pioneering "0-to-1" benchmark for assessing cultural awareness of reward models, we prioritized cultural evaluation data validity and the availability of qualified native annotators over broad but potentially unreliable culture coverage. Focusing on these representative high/medium-resource cultures allowed us to ensure strict quality control through native verification (as detailed in **Response to W-1 & W-2**). We believe this rigorous foundation is the necessary first step before expanding to lower-resource regions in future work.
>
> > Q-1 Comment: Values in Fig. 3 are a bit difficult to see, consider using a darker colour for the grid.
> >
>
> Thank you for the helpful suggestion regarding the presentation of our results. Your constructive feedback will further enhance the readability and overall clarity of this work.
>
> **Action taken**: In the revised manuscript, the grid color in **Figure 3** has been darkened for better readability.

---

> > ### Comment · Reviewer_9wq4 · 2025-11-24
> >
> > Thank you very much for the additional results and clarifications.
> > I think adding a human-authored validation set is important for objectively measuring progress.
> > How does the proposed RM perform on the new human-authored validation set?

---

> ### Author Response · Authors · 2025-11-25
> **Response to Reviewer 9wq4**
>
> We express gratitude for this constructive comment and for engaging further in the discussion.
>
> > Additional Comment: Thank you very much for the additional results and clarifications. I think adding a human-authored validation set is important for objectively measuring progress. How does the proposed RM perform on the new human-authored validation set?
> >
>
> In our previous response to W-1 and W-2, we have provided a new human-authored validation set evaluation in the current SOTA reward models, and then, following your constructive comment, we evaluated our proposed RM on the newly constructed human-authored CARB validation set, alongside all baseline reward models. The full results are shown in the following table and included in the revised manuscript.
>
> | **Models** | **Original Set (CARB)** |  |  |  | **Human-Written Set (CARB)** |  |  |  |
> | --- | --- | --- | --- | --- | --- | --- | --- | --- |
> |  | Arabic | Chinese | Japanese | **Avg.** | Arabic | Chinese | Japanese | **Avg.** |
> | ***Classifier-based RMs*** |  |  |  |  |  |  |  |  |
> | Skywork-Reward-Gemma-2-27B | 67.6 | 74.9 | 76.6 | 72.6 | 57.0 | 58.0 | 53.0 | 56.0 |
> | INF-ORM-Llama3.1-70B | 67.6 | 71.2 | 74.2 | 70.7 | 49.0 | 61.0 | 52.0 | 54.0 |
> | QRM-Gemma-2-27B | 63.4 | 67.3 | 77.8 | 69.1 | 54.0 | 55.0 | 50.0 | 53.0 |
> | Llama-3.1-70B-Instruct-RM-RB2 | 63.9 | 70.3 | 72.8 | 68.6 | 55.0 | 56.0 | 48.0 | 53.0 |
> | ***Generative RMs*** |  |  |  |  |  |  |  |  |
> | Qwen3-235B-A22B-Instruct | 69.9 | 81.4 | 78.1 | 76.0 | 58.0 | 63.0 | 58.0 | 59.7 |
> | DeepSeek-V3-0324 | 68.5 | 80.7 | 74.6 | 74.2 | 56.0 | 66.0 | 58.0 | 60.0 |
> | GPT-4o-0806 | 67.6 | 73.6 | 76.6 | 72.3 | 54.0 | 54.0 | 50.0 | 52.7 |
> | Qwen2.5-7B-Instruct | 54.7 | 70.6 | 64.4 | 62.6 | 49.0 | 52.0 | 44.0 | 48.3 |
> | ***Reasoning RMs*** |  |  |  |  |  |  |  |  |
> | DeepSeek-GRM-27B | 53.2 | 62.8 | 63.7 | 59.9 | 54.0 | 55.0 | 45.0 | 51.3 |
> | JudgeLRM-7B | 50.5 | 61.4 | 58.6 | 56.8 | 47.0 | 50.0 | 41.0 | 46.0 |
> | RM-R1-Qwen-Instruct-7B | 46.6 | 62.3 | 54.9 | 54.6 | 42.0 | 52.0 | 38.0 | 44.0 |
> | RM-R1-Qwen-Instruct-7B\textsuperscript{\textdagger} | 67.8 | 81.8 | 77.0 | 75.5 | 58.0 | 63.0 | 55.0 | 58.7 |
> | **Think-as-Locals (based on Qwen2.5-7B-Instruct)** | 72.2 | 82.9 | 81.2 | **78.8** | **64.0** | **65.0** | **59.0** | **62.7** |
>
> On the original CARB validation set, our RM achieves the best average accuracy (78.8), outperforming all classifier-based, generative, and reasoning RMs. On the new human-authored validation set—which is clearly more challenging, as all models’ scores drop—**our RM (Think-as-Locals) again achieves the highest average accuracy (62.7)**, consistently outperforming the strongest baselines (e.g., Qwen3-235B-A22B-Instruct at 59.7 and DeepSeek-V3-0324 at 60.0).
>
> We also observe that, despite the overall decrease in absolute performance, the relative ranking of models is largely preserved between the original and human-authored sets. This supports **the validity of the original CARB benchmark while showing that our method remains effective under a more realistic, human-authored evaluation**. We will release the human-authored validation set together with the original CARB data to better support objective progress measurement for future work.

---

> > ### Comment · Reviewer_9wq4 · 2025-11-25
> >
> > Thank you very much for the additional results. I believe this comparison is crucial for more accurately reflecting the true progress of cultural awareness in RMs. I have updated my scores, and I hope the authors incorporate all the additional results into the revised version of the paper.

---

> ### Author Response · Authors · 2025-11-25
> **Sincerely appreciate for your valuable feedback!**
>
> Dear Reviewer 9wq4,
>
> Thank you so much again for your comprehensive and constructive review!
>
> All of your questions and comments were incredibly insightful and constructive in improving the quality of our manuscript. We have incorporated all the new experimental analyses during the rebuttal period into the manuscript, and we commit to ensuring that these updates will also be included in the final version. The revisions we made based on your feedback have truly sharpened the core contributions and clarity of our work.
>
> We once again sincerely appreciate the valuable time and expertise you dedicated to our paper.
>
> With deepest thanks,
>
> All authors

---

### Official Review · Reviewer_SHu3 · 2025-10-29

**Soundness:** 2
**Presentation:** 3
**Contribution:** 3
**Rating:** 6
**Confidence:** 4

**Summary:**

This paper addresses an under explored yet crucial dimension of aligning large language models (LLMs): cultural awareness in reward modeling. The authors first diagnose the shortcomings of existing reward model (RM) evaluations, which mainly focus on general capabilities and lack culturally grounded benchmarks. To bridge this gap, they introduce CARB (Cultural Awareness Reward modeling Benchmark), covering 10 cultures across 4 cultural domains. The authors then show that CARB scores correlate strongly with downstream multilingual cultural alignment tasks, based on which the paper claims CARB is effective as an evaluation benchmark for RM. Subsequently, the authors show that current RMs struggle to model authentic cultural nuances, often relying on surface-level correlations rather than meaningful cultural reasoning.
To mitigate the identified issues, the authors finally propose Think-as-Locals, a reinforcement learning from verifiable rewards (RLVR) approach, designed to elicit deeper, culture-aware reasoning and reduce spurious feature reliance.

**Strengths:**

- The paper is well written and easy to follow, with a clear logical flow: each section identifies a limitation and proposes a corresponding solution.

- The topic is timely and important, addressing cultural inclusivity/awareness, a crucial but underexplored aspect of global AI alignment.

- The strategy "Think-as-Locals" seems to be an effective RLVR method to elicit deep cultural understanding in generative RMs.

- The experiments are quite extensive with large-scale multilingual evaluation and ablation studies.

**Weaknesses:**

**Major Weaknesses**

(W1) The authors use GPT-4o to translate English prompts into other languages, and report that three human annotators manually refined them. However, according to Section B.1, these annotators are independent undergraduate and graduate students. However, the authors do not report if the human annotators are native speakers (or even fluent speakers) of the studied languages. This raises concerns about the quality and cultural authenticity of the prompts.

(W2) The same concern applies to the human annotation agreement results. If the annotators are not from the respective cultural backgrounds, the evaluation may not fully reflect genuine cultural preferences. A more rigorous evaluation involving native speakers/members of the cultural regions would increase the reliability of CARB and better ensure that it captures authentic human judgments.

(W3) In Section 5, GPT-4o is used to rate generated responses for cultural relevance, faithfulness, and helpfulness. These are complex subjective dimensions, and it is debatable whether GPT-4o can serve as a reliable judge for such evaluations. Even if using LLM-as-a-Judge might be necessary for an efficient evaluation, the authors should provide a human meta evaluation of the LLM-judge in the different cultural settings.

(W3) While the Think-as-Locals method appears effective, it likely increases computational cost, as rewards depend on the full reasoning traces rather than final outputs.


**Minor Weaknesses**

- The selected set of cultures is essential to this work and its list should not be hidden in the Appendix (Table 4).

- While CARB covers 10 cultures, its cultural coverage remains limited compared to the global landscape; especially with regard to lower resource languages. Extending the benchmark to more low-resource or underrepresented regions (e.g., from Africa) would enhance representativeness.

- Figure 5 shows that the linear relationship is weak and statistically insignificant for M-RewardBench but strong for CARB. The paper should provide a clearer explanation or hypothesis for this difference.

- Including an analysis of failure cases (e.g., where Think-as-Locals still misinterprets cultural cues) would enhance the understanding of this approach.

- Cultural evaluation is a rapidly growing area and the authors list a lot of relevant works. Some additional, recent works could also be relevant to their section on related work: https://aclanthology.org/2024.acl-long.345/ https://aclanthology.org/2024.acl-long.862/ https://aclanthology.org/2025.naacl-long.402/ https://arxiv.org/abs/2505.21693

- The proposed reward R_{appr} is claimed to be motivated by intrinsic probability, but no intuitive explanation is given. A brief intuition behind this design would improve understanding.

- The authors mention "CARE" in line 275. I first thought this was a typo (of their "CARB"). But actually, it is a different dataset which is only properly introduced with reference in line 408.

**Typos **

Line 143: structured. -> structured, (comma instead of full stop)

**Questions:**

- How scalable is CARB in practice? Can it be easily extended to new cultural domains or low-resource languages with limited data?

- How does CARB perform when applied to unseen cultures or languages in post-training or adaptation scenarios?

- Did the human annotators have a suitable background to be able to judge and annotate the diverse cultural regions? (see W2)

- Does the LLM-as-a-Judge align with human judgments? (see W3)

- How does the training time of the more complex procss compares to using a standard reward computed on the final answer only? (see W4)

---

> ### Author Response · Authors · 2025-11-21
> **Response to Reviewer SHu3 (1/5)**
>
> Thank you sincerely for your thoughtful and positive feedback on our work. Below, we have provided a detailed explanation for your concern as follows. In accordance with your suggestion, we have made revisions to our manuscripts. Major updates include the addition of native-level verification (**Section 3; Appendices B.3, D.1**), a latency analysis of the proposed method (**Appendix G.5**), a failure case study (**Appendix G.6**), and an evaluation of the method’s generalization ability to unseen cultures (**Appendix G.3**). We have also improved the writing by clarifying the selection of cultures in **Section 3**, adding an explanation of the correlation analysis in **Section 5**, enriching the intuitive explanation of the proposed methodology in **Section 7**, and updating the related work in **Section 2**. All styling issues and **typographical** **errors have been corrected**. All revisions made to the paper are highlighted in blue for your ease of reference. Please do not hesitate to let us know if you have any further questions.
>
> ---
>
> > W-1 The authors use GPT-4o to translate English prompts into other languages, and report that three human annotators manually refined them. However, according to Section B.1, these annotators are independent undergraduate and graduate students. However, the authors do not report if the human annotators are native speakers (or even fluent speakers) of the studied languages. This raises concerns about the quality and cultural authenticity of the prompts.
> >
>
> Thank you for pointing out this important consideration. Your feedback is highly valuable and provides meaningful direction for the continued development and improvement of our benchmark. We wish to address the concern regarding cultural authenticity by clarifying our original workflow and providing new empirical validation.
>
> - **Clarification of expert oversight in original process:** Our original refinement process (**Sec. 3, App. B.1, B.3**) extended beyond the three human annotators. Crucially, it was supervised by two professional linguists from Lan-bridge, an ISO-recognized translation provider who are proficient in most of the ten cultures. These experts acted as instructors and assessors, ensuring that the prompts met **professional, native-level standards across the ten covered cultures**.
> - **Native speaker validation:** To explicitly and empirically address concerns regarding native-level validation, we conducted an additional evaluation study during the rebuttal period. **We recruited 10 new native-speaker evaluators—overseas students currently studying at Xi’an International Studies University**, each representing one cultural group and holding at least an undergraduate degree. These evaluators assessed the cultural appropriateness and quality (whether exhibiting translationese or western-centric phrasing) of our refined prompts using 100 randomly selected samples per language (25 for each cultural dimension), resulting in 1,000 samples in total. This procedure ensured that only prompts judged to be both high-quality and culturally appropriate were retained.
>
>     **Results:** As detailed in the table below, the native evaluators reported an average **94.7% agreement rate** on our refined prompts. This high agreement rate confirms that **our expert-guided refinement process successfully mitigated translationese or western-centric phrasing and achieved native-level quality across all cultures**.
>
>     | Language | Prompt Agreement (%) |
>     | --- | --- |
>     | Spanish | 96.0 |
>     | German | 92.0 |
>     | Thai | 98.0 |
>     | Vietnamese | 99.0 |
>     | Korean | 90.0 |
>     | Chinese | 98.0 |
>     | Arabic | 90.0 |
>     | Russian | 91.0 |
>     | English | 98.0 |
>     | Japanese | 95.0 |
>
> **Action taken**: We have revised the manuscript to include this native-level verification experiment in **Section 3** and details in **Appendix B.3** to highlight the strong correlation between our refinement process and native-level standards.
>
> ---

---

> ### Author Response · Authors · 2025-11-21
> **Response to Reviewer SHu3 (2/5)**
>
> > W-2 The same concern applies to the human annotation agreement results. If the annotators are not from the respective cultural backgrounds, the evaluation may not fully reflect genuine cultural preferences. A more rigorous evaluation involving native speakers/members of the cultural regions would increase the reliability of CARB and better ensure that it captures authentic human judgments.
> >
>
> > Q-3 Did the human annotators have a suitable background to be able to judge and annotate the diverse cultural regions? (see W2)
> >
>
> As mentioned in the response to W-1, we wish to clarify that our original annotation process, as detailed in Sec. 3 and Appendix B.3, involved three annotators *under the guidance of two professional translation experts* proficient in most of the ten cultures to ensure the authentic native-level review.
>
> **Native speaker validation**: To more directly address concerns regarding cultural contrast and the accuracy of our overall construction process (embedding similarity combined with GPT-4o judgments), we conducted a new native-speaker evaluation during the rebuttal period, using the same annotators referenced in our response to W-1. These annotators assessed cultural alignment for accepted completions and cultural inaccuracies for rejected ones across 1,000 randomly selected samples—100 samples per language, each containing 25 instances representing every cultural dimension.
>
> **Results**: The results in the following table show that **native annotators achieved a high agreement ratio with our automated construction method**. The high agreement rates verify that **our constructions align closely with native-level cultural preference standards**, thereby directly addressing the concern.
>
> | Language | Chosen Completion Agreement (%) | Rejected Completions Agreement (%) |
> | --- | --- | --- |
> | Spanish | 94.0 | 92.3 |
> | German | 91.0 | 91.0 |
> | Thai | 99.0 | 93.3 |
> | Vietnamese | 99.0 | 99.3 |
> | Korean | 87.0 | 84.0 |
> | Chinese | 93.0 | 90.3 |
> | Arabic | 84.0 | 77.0 |
> | Russian | 84.0 | 86.7 |
> | English | 95.0 | 93.7 |
> | Japanese | 93.0 | 80.0 |
>
> **Action taken**: We have revised the manuscript to include this native-level annotation experiment in **Section 3** and details in **Appendix B.3** to highlight the strong agreement between our construction process and native-level annotators.
>
> ---
>
> > W-3 In Section 5, GPT-4o is used to rate generated responses for cultural relevance, faithfulness, and helpfulness. These are complex subjective dimensions, and it is debatable whether GPT-4o can serve as a reliable judge for such evaluations. Even if using LLM-as-a-Judge might be necessary for an efficient evaluation, the authors should provide a human meta evaluation of the LLM-judge in the different cultural settings.
> >
>
> > Q-4 Does the LLM-as-a-Judge align with human judgments? (see W3)
> >
>
> We appreciate the reviewer’s emphasis on evaluation reliability. Actually, as stated in **Lines 248-249**, we followed established work [1] using LLM-as-Judge to evaluate cultural relevance. Additionally, as noted in **Appendix D.1 (Lines 2286-2289)**, our submission already includes a human meta-evaluation supporting the reliability of the LLM judge.
>
> To further quantify this alignment, we conducted an additional analysis using the open-source CARE dataset [1]. The resulting Kendall coefficients indicate a strong correlation between GPT-4o and native human judgments:
>
> | Kendall coefficient | Arabic | Chinese | Japanese |
> | --- | --- | --- | --- |
> | GPT-4o-as-a-Judge vs. Human | 0.508 | 0.659 | 0.657 |
>
> **Collectively, these results demonstrate that GPT-4o's evaluations align well with native human cultural preferences in complex subjective dimensions, supporting its reliability for our study.**
>
> **Action taken**: We have revised the manuscript to include a cross-reference in **Section 5** and include this additional correlation experiment in **Appendix D.1** to highlight the strong alignment between LLM-as-a-Judge and human judgments.
>
> ---

---

> ### Author Response · Authors · 2025-11-21
> **Response to Reviewer SHu3 (3/5)**
>
> > W-4 While the Think-as-Locals method appears effective, it likely increases computational cost, as rewards depend on the full reasoning traces rather than final outputs.
> >
>
> > Q-5 How does the training time of the more complex procss compares to using a standard reward computed on the final answer only? (see W4)
> >
>
> We acknowledge the reviewer's concern regarding the computational overhead of the "Think-as-Locals" method. To address this, we have provided a detailed comparison of the training time efficiency in **Figure 31** in the revised manuscript.
>
> Our training experiments were conducted on a computational node equipped with 8 $\times$ NVIDIA H20 GPUs (96GB VRAM each). We utilized the Qwen2.5-Instruct-7B as the base model, training with a learning rate of $1.0 \times 10^{-6}$ using the verl framework [2].
>
> As illustrated in **Figure 31**:
>
> - **Baseline (Red Line):** The standard Correctness Reward method, which computes rewards solely on the final answer, averages approximately **530 seconds per step**.
> - **Our Method (Blue Line):** The proposed method, which incorporates token-level probability rewards over the full reasoning traces, averages approximately **700 seconds per step**.
>
> While our method introduces a per-step latency increase of approximately 32% (an addition of $\sim170$ seconds/step), this is an expected trade-off for the denser supervisory signal provided by the reasoning traces. We mitigate potential bottlenecks by caching intermediate log-likelihood computations where possible. We find this overhead to be within acceptable limits for the significant performance gains achieved in cultural alignment.
>
> **Action taken**: We have revised the manuscript to include the training time latency analysis into the **Appendix G.5.**
>
> ---
>
> > W-5 The selected set of cultures is essential to this work and its list should not be hidden in the Appendix (Table 4).
> >
>
> Thank you for your kind suggestion. We agree with the reviewer on the importance of the selected cultures. This list was initially placed in the appendix to adhere to the submission's strict page limits. We have added selected cultures to **Section 3** in the revised manuscript.
>
> **Action taken**: We have revised the manuscript to include the list of selected cultures in **Section 3**.
>
> ---
>
> > W-6 While CARB covers 10 cultures, its cultural coverage remains limited compared to the global landscape; especially with regard to lower resource languages. Extending the benchmark to more low-resource or underrepresented regions (e.g., from Africa) would enhance representativeness.
> >
>
> We appreciate the reviewer acknowledging the substantial challenge of rigorously assessing 10 cultures. As a pioneering "0-to-1" benchmark for assessing cultural awareness of reward models, we prioritized cultural evaluation data validity and the availability of qualified native annotators over broad but potentially unreliable culture coverage. Focusing on these representative high/medium-resource cultures allowed us to ensure strict quality control through native verification (as detailed in **Response to W-1 & W-2**). We believe this rigorous foundation is the necessary first step before expanding to lower-resource regions in future work.
>
> ---
>
> > W-7 Figure 5 shows that the linear relationship is weak and statistically insignificant for M-RewardBench but strong for CARB. The paper should provide a clearer explanation or hypothesis for this difference.
> >
>
> We sincerely appreciate the reviewer’s kind suggestion. We detail the explanation of this difference as follows:
>
> The weak and statistically insignificant correlation for M-RewardBench is key evidence demonstrating that this **existing benchmark fails to effectively measure a reward model's cultural awareness**. In contrast, the strong correlation observed with our proposed CARB provides direct validation that it does successfully assess this capability. **This result substantiates our claim that CARB is a more effective and necessary benchmark for evaluating the cultural awareness of reward models**, which is the primary research problem we address.
>
> **Action taken**: We have revised the manuscript to include a clearer explanation for this difference in **Section 5.2**.
>
> ---

---

> ### Author Response · Authors · 2025-11-21
> **Response to Reviewer SHu3 (4/5)**
>
> > W-8 Including an analysis of failure cases (e.g., where Think-as-Locals still misinterprets cultural cues) would enhance the understanding of this approach.
> >
>
> Thanks for this constructive suggestion. Actually, we have already included a success case study of the proposed Think-as-Locals method in **Section G.3**. And to provide a balanced perspective, we analyzed representative failure cases where Think-as-Locals diverges from gold references. These instances reveal minor, nuanced limitations in handling high-ambiguity cultural contexts:
>
> - *Nuanced Idiomatic Alignment***:** Occasional difficulty in selecting the optimal trade-off between literal accuracy and culturally equivalent metaphors (e.g., mapping specific proverbs). (Failure case in **Table 33**)
> - *Intra-Cultural Heterogeneity***:** Challenges in navigating topics where local values are not monolithic, such as balancing traditional filial piety with modern individualism. (Failure case in **Table 34**)
> - *Context-Dependent Norms***:** Minor misjudgments regarding social etiquette that fluctuate based on specific settings (e.g., interpreting noise levels as hospitality versus impoliteness). (Failure case in **Table 35**)
>
> These cases primarily reflect the inherent complexity and subjective nature of deep cultural interpretation rather than fundamental reasoning errors, and **do not undermine the model's overall effectiveness** in alleviating the spurious correlation in cultural reward modeling.
>
> **Action taken:** We have revised the manuscript to include the failure case study in **Section G.6**.
>
> ---
>
> > W-9 Cultural evaluation is a rapidly growing area and the authors list a lot of relevant works. Some additional, recent works could also be relevant to their section on related work: https://aclanthology.org/2024.acl-long.345/ https://aclanthology.org/2024.acl-long.862/ https://aclanthology.org/2025.naacl-long.402/ https://arxiv.org/abs/2505.21693
> >
>
> Thank you for identifying these recent works. We have incorporated all four into our revised related work section:
>
> Wang et al.[3] identify cultural dominance issues in LLMs; Naous et al.[4] propose CAMEL, a resource for measuring cultural biases through extrinsic and intrinsic evaluations; Nikandrou et al.[5] introduce a visual question answering benchmark to probe culture-specific concept knowledge and cultural adaptation capacity; and Zhao et al.[6] propose an automatic Wikidata-based multilingual framework for evaluating cultural awareness across languages, regions, and topics.
>
> **Action taken**: We have revised the manuscript to include this supplementary related work in **Section 2**.
>
> ---
>
> > W-10 The proposed reward R_{appr} is claimed to be motivated by intrinsic probability, but no intuitive explanation is given. A brief intuition behind this design would improve understanding.
> >
>
> The core intuition behind $\mathcal{R}_{appr}$ is to **quantify how much the generated cultural reasoning (the criteria $z$) causally contributes to reaching the correct final judgment ($j$)**. We want to reward reasoning chains that make the *correct* answer *more probable* to the model.
>
> As we state in the paper, our design is motivated by the finding that LLMs can evaluate reasoning quality via their own intrinsic probability. Our $\mathcal{R}_{appr}$ formula is a direct implementation of this idea:
>
> 1. It first calculates the probability of the ground-truth judgment $j$ *given* the generated reasoning $z$ (the $r_{\theta}(z_{t}^{\prime}|...)$ term).
> 2. It then subtracts the baseline probability of the ground-truth judgment $j$ *without* that specific reasoning (the $r_{\theta}(j|...)$ term).
>
> Thus, $\mathcal{R}_{appr}$ isolates the **net "probability boost"** provided by the reasoning $z$ itself. A high reward indicates the reasoning was appropriate and helpful for arriving at the correct cultural judgment, while a low or negative reward indicates the reasoning was irrelevant or misleading.
>
> **Action taken**: We have revised the manuscript to include this intuition to strengthen the clarity and insight in **Section 7**.
>
> ---
>
> > W-11 The authors mention "CARE" in line 275. I first thought this was a typo (of their "CARB"). But actually, it is a different dataset which is only properly introduced with reference in line 408.
> >
>
> > W-12 *Typos ** Line 143: structured. -> structured, (comma instead of full stop)
> >
>
> We sincerely appreciate the reviewer for the careful reading and for catching these typos.
>
> **Action taken**:
>
> - Regarding **W-11**, the reviewer is correct: the mention of "CARE" in line 275 was a typo and should be our proposed "CARB". We have corrected this in the revision.
> - Regarding **W-12**, we have corrected the punctuation in **Line 143**.
>
> We also conduct a thorough proofreading of the entire manuscript to resolve any remaining typos and grammatical errors.
>
> ---

---

> ### Author Response · Authors · 2025-11-21
> **Response to Reviewer SHu3 (5/5)**
>
> > Q-1 How scalable is CARB in practice? Can it be easily extended to new cultural domains or low-resource languages with limited data?
> >
>
> **CARB is highly scalable and designed for extensibility.** As detailed in **Section 3**, our construction framework is methodologically agnostic to specific cultural domains. We empirically demonstrated this adaptability by successfully generating 800 Best-of-N test sets specifically for limited-data language settings. Consequently, extending CARB to new low-resource domains is technically straightforward; the practical requirement for native speaker verification is a deliberate design choice to ensure cultural authenticity, not an inherent limitation of the framework itself. Therefore, while our *framework* is scalable, its practical *application* to new low-resource domains is contingent on overcoming this external data-sourcing and expert-validation challenge, which is a significant practical hurdle common to the field.
>
> ---
>
> > Q-2 How does CARB perform when applied to unseen cultures or languages in post-training or adaptation scenarios?
> >
>
> I wonder if the reviewer wants to ask about the proposed method Think-as-Locals when it is applied to unseen cultures?
>
> If so, we evaluated our proposed method (trained on Chinese, Japanese, and Arabic) on three **unseen cultures**: Korean, Spanish, and Vietnamese. The results below show that our proposed method, while trained on several cultures, can generalize well to unseen cultures. We attribute this generalization to the **paradigm shift** introduced by our method. By transitioning from a discriminative judgment approach to a **generative Chain-of-Thought (CoT) reasoning paradigm**, the model learns a transferable reasoning structure rather than merely fitting to specific language statistics. This allows the model to apply "local perspective-taking" strategies even when applied to unseen cultural contexts.
>
> | Train on Chinese, Japanese, and Arabic | M-Rewardbench (seen cultures) | CARB (seen cultures) | M-Rewardbench (unseen cultures) | CARB (unseen cultures) |
> | --- | --- | --- | --- | --- |
> | Qwen-2.5-7B-Instruct | 77.1 | 62.6 | 76.6 | 62.2 |
> | Think-as-Locals-7B | 80.4 | 78.8 | 79.0 | 77.3 |
>
> **If I have misunderstood your question, could you kindly let me know. I would be glad to continue the discussion in more detail.**
>
> **Action taken**: We have revised the manuscript to include the out-of-domain evaluation in **Section 7** and **Appendix G.3**.
>
> ---
>
> **Reference**
>
> [1] Guo, G., Naous, T., Wakaki, H., Nishimura, Y., Mitsufuji, Y., Ritter, A. and Xu, W., 2025, November. CARE: Multilingual Human Preference Learning for Cultural Awareness. In *Proceedings of the 2025 Conference on Empirical Methods in Natural Language Processing* (pp. 32854-32883).
>
> [2] Sheng, G., Zhang, C., Ye, Z., Wu, X., Zhang, W., Zhang, R., Peng, Y., Lin, H. and Wu, C., 2025, March. Hybridflow: A flexible and efficient rlhf framework. In *Proceedings of the Twentieth European Conference on Computer Systems* (pp. 1279-1297).
>
> [3] Wang, W., Jiao, W., Huang, J., Dai, R., Huang, J.T., Tu, Z. and Lyu, M., 2024, August. Not all countries celebrate thanksgiving: On the cultural dominance in large language models. In *Proceedings of the 62nd Annual Meeting of the Association for Computational Linguistics (Volume 1: Long Papers)* (pp. 6349-6384).
>
> [4] Naous, T., Ryan, M.J., Ritter, A. and Xu, W., 2024, August. Having beer after prayer? measuring cultural bias in large language models. In *Proceedings of the 62nd Annual Meeting of the Association for Computational Linguistics (Volume 1: Long Papers)* (pp. 16366-16393).
>
> [5] Nikandrou, M., Pantazopoulos, G., Vitsakis, N., Konstas, I. and Suglia, A., 2025, April. CROPE: Evaluating in-context adaptation of vision and language models to culture-specific concepts. In *Proceedings of the 2025 Conference of the Nations of the Americas Chapter of the Association for Computational Linguistics: Human Language Technologies (Volume 1: Long Papers)* (pp. 7917-7936).
>
> [6] Zhao, R., Chen, B., Plank, B. and Hedderich, M.A., 2025. MAKIEval: A Multilingual Automatic WiKidata-based Framework for Cultural Awareness Evaluation for LLMs. *arXiv preprint arXiv:2505.21693*.

---

> > ### Comment · Reviewer_SHu3 · 2025-11-26
> >
> > Thank you for your effort in adressing my concerns in such detail. Most of my issues were resolved and I have updated the scores accordingly.

---

> > > ### Author Response · Authors · 2025-11-26
> > > **Sincerely appreciate for your valuable feedback!**
> > >
> > > Dear Reviewer SHu3,
> > >
> > > Thank you very much for your thorough and constructive review.
> > >
> > > We sincerely appreciate the valuable time and expertise you have devoted to evaluating our paper. Your questions and comments were exceptionally insightful and greatly contributed to improving the quality of our manuscript. They have not only helped sharpen the core contributions and enhance the clarity of our work, but also inspired us to consider new directions for future exploration.
> > >
> > > We are also pleased to hear that your concerns have been addressed, and we hope the revised version meets your expectations. Should you have any further suggestions, we’re always open to further discussion :-)
> > >
> > > With our deepest gratitude,
> > >
> > > All authors

---

### Official Review · Reviewer_ZBoR · 2025-10-30

**Soundness:** 2
**Presentation:** 2
**Contribution:** 3
**Rating:** 4
**Confidence:** 4

**Summary:**

This paper introduces CARB (Cultural Awareness Reward modeling Benchmark), a benchmark designed to evaluate and improve the cultural awareness of reward models (RMs) used in aligning large language models (LLMs). CARB includes 10 cultures and 4 domains covering cultural commonsense, values, safety, and linguistics, using a Best-of-N evaluation on human-curated prompts and LLM-generated responses. Results show that current RMs perform inconsistently across cultures, often relying on spurious correlations such as surface linguistic features rather than genuine cultural understanding, though CARB scores correlate positively with downstream multilingual cultural alignment performance. To address these limitations, the paper also introduces Think-as-Locals, an RLVR-based framework that encourages RMs to explicitly generate culturally grounded evaluation criteria before judgment. Experiments demonstrate that Think-as-Locals mitigates spurious biases and enhances the cultural sensitivity and reasoning depth of reward models.

**Strengths:**

**S1.** CARB covers 10 cultures with 4 cultural domains, and the final dataset contains 8,576 prompts with 24 LLM-generated responses, which is very diverse.

**S2.** The study involves large-scale and comprehensive evaluations of a lot of different reward models (such as classifier-based and generative reward models), cross-lingual comparisons, and correlation studies with downstream multilingual alignment tasks.

**S3.** The paper introduces new framework called Think-as-Locals that is shown to encourage culturally grounded reasoning and effectively mitigates spurious correlations in reward modeling, as it outperform strong baselines across multilingual cultural benchmarks, even compared to other method with retraining (i.e. RM-R1).

**Weaknesses:**

**W1.** Here I group all weaknesses related to CARB as a dataset:
- CARB relies heavily on GPT-4o for prompt generation and translation, which may introduce translationese or Western-centric phrasing [1]. While three human annotators refined the prompts, the dataset spans ten cultures, which makes me assume that only a subset of languages or cultures received native-level review.
- The construction of “rejected” responses using embedding similarity and the validation of human annotations with GPT-4 risk circular reasoning, as embedding distance and model agreement do not necessarily capture true cultural contrast or correctness.
- During generation of the dataset (Figure 14), it appears to assume a one-to-one correspondence between language and culture, which would overlook multilingual or multicultural contexts where a single language represents diverse cultural norms.
- Some cultural values and commonsense knowledge evolve over time, which potentially misrepresenting shifting cultural norms.
- Finally, the cultures being considered are limited to languages that are relative high/medium-resourced languages. However, I would say 10 cultures/languages are quite a lot already if the dataset is assessed by native people.

**W2.** Here I group all weaknesses related to Multilingual Alignment Performance evaluation:
- While I agree that m-reward-bench does not provide multicultural evaluation, I don't think correlation of performance between one benchmark and another is a good indicator of a well-correlated benchmark in terms of cultural knowledge, as there could be many confounding factors that represent such correlation. In addition, it only measures linear correlation, doesn't necessarily mean there is no correlation, furthermore in terms of causality that CARB represents multicultural benchmark.

Suppose correlation were a good measure. Notice that include-base-44 itself consists of academic and professional exams, which also includes cultural-agnostic questions, AlpacaEval (which is the source of OMGEval) contain a lot of questions that are not culturally related (as shown in small number of localizations in the original paper), and BLEND's answers depend on the annotations being given (not representative of the culture of a country as a whole), so another reliance on GPT-4 as a judge wouldn't be great. If correlation were a good measure, this wouldn't remove the possibility that the performance could be correlated to the cultural-agnostic questions.

- Section 5 claims strong “multilingual” correlation between CARB and downstream alignment, yet Section 6.2 explicitly shows that reward models exhibit cross-lingual inconsistency by assigning divergent scores to the same semantic content across languages. Wouldn't this show that CARB’s predictive validity about assessing non-shallow cultural features to be wrong?

**W3.** While the result for the reward model framework looks promising on m-reward-bench and CARB, there is a lack of evidence that the reward model can be used to successfully post-train a policy model (RLHF or DPO).

## References

[1] Yan, J., Yan, P., Chen, Y., Li, J., Zhu, X., & Zhang, Y. (2024). Gpt-4 vs. human translators: A comprehensive evaluation of translation quality across languages, domains, and expertise levels. arXiv preprint arXiv:2407.03658.

**Questions:**

Beyond my comments in weaknesses, here are my questions:

**Q1.** Related to **W1**, what kind of metric did you use to measure the inter-annotator agreement between GPT-4o since it's in percentage?

**Q2.** Most of the reward models used are LLM-as-a-judge(s) or reward model that is trained in English, how about specific multilingual models that (possibly) learn cultures during training (from recent related multilingual works), such as mR3 [1], Multilingual Nemotron [2], m-prometheus [3]?

**Q3.** After RLVF, seems like there is much more improvement in CARB compared to m-reward-bench, and I wonder if it's due to in-domain evaluation as the curated training dataset also involves CARB?

## References

[1] Anugraha, D., Hung, S. Y., Tang, Z., Lee, A. E. S., Wijaya, D. T., & Winata, G. I. (2025). mR3: Multilingual Rubric-Agnostic Reward Reasoning Models. arXiv preprint arXiv:2510.01146.

[2] Wang, Z., Zeng, J., Delalleau, O., Egert, D., Evans, E., Shin, H. C., ... & Kuchaiev, O. (2025). HelpSteer3: Human-Annotated Feedback and Edit Data to Empower Inference-Time Scaling in Open-Ended General-Domain Tasks. arXiv preprint arXiv:2503.04378.

[3] Pombal, J., Yoon, D., Fernandes, P., Wu, I., Kim, S., Rei, R., ... & Martins, A. F. (2025). M-Prometheus: A Suite of Open Multilingual LLM Judges. arXiv preprint arXiv:2504.04953.

---

> ### Author Response · Authors · 2025-11-21
> **Response to Reviewer ZBoR (1/6)**
>
> We sincerely thank the reviewer for providing valuable comments. Below, we provide a detailed point-by-point response. Corresponding revisions in the manuscript have been highlighted in blue for reference. Major updates include the addition of native-level verification (**Section 3, Appendix B.3**), the proposed method benefits LLM alignment (**Section 7)**, expanded evaluation results for multilingual models (**Table 3**), and further controlled validation experiments (**Appendices B.1, D.6, G.4**). Additionally, we have refined phrasing to **reduce ambiguity** and **corrected all styling errors and typos**. Please kindly let us know if you have any further concerns.
>
> ---
>
> > W-1 CARB relies heavily on GPT-4o for prompt generation and translation, which may introduce translationese or Western-centric phrasing [1]. While three human annotators refined the prompts, the dataset spans ten cultures, which makes me assume that only a subset of languages or cultures received native-level review.
> >
>
> Thank you for pointing out this important consideration. Your feedback is highly valuable and provides meaningful direction for the continued development and improvement of our benchmark. We wish to address the concern regarding cultural authenticity by clarifying our original workflow and providing new empirical validation.
>
> - **Clarification of expert oversight in original process:** Our original refinement process (**Sec. 3, App. B.1, B.3**) extended beyond the three human annotators. Crucially, it was supervised by two professional linguists from Lan-bridge, an ISO-recognized translation provider who are proficient in most of the ten cultures. These experts acted as instructors and assessors, ensuring that the prompts met **professional, native-level standards across the ten covered cultures**.
> - **Native speaker validation:** To explicitly and empirically address concerns regarding native-level validation, we conducted an additional evaluation study during the rebuttal period. **We recruited 10 new native-speaker evaluators—overseas students currently studying at Xi’an International Studies University**, each representing one cultural group and holding at least an undergraduate degree. These evaluators assessed the cultural appropriateness and quality (whether exhibiting translationese or western-centric phrasing) of our refined prompts using 100 randomly selected samples per language (25 for each cultural dimension), resulting in 1,000 samples in total. This procedure ensured that only prompts judged to be both high-quality and culturally appropriate were retained.
>
>     **Results:** As detailed in the table below, the native evaluators reported an average **94.7% agreement rate** on our refined prompts. This high agreement rate confirms that **our expert-guided refinement process successfully mitigated translationese or western-centric phrasing and achieved native-level quality across all cultures**.
>
>     | Language | Prompt Agreement (%) |
>     | --- | --- |
>     | Spanish | 96.0 |
>     | German | 92.0 |
>     | Thai | 98.0 |
>     | Vietnamese | 99.0 |
>     | Korean | 90.0 |
>     | Chinese | 98.0 |
>     | Arabic | 90.0 |
>     | Russian | 91.0 |
>     | English | 98.0 |
>     | Japanese | 95.0 |
>
> **Action taken**: We have revised the manuscript to include this native-level verification experiment in **Section 3** and details in **Appendix B.3** to highlight the strong correlation between our refinement process and native-level standards.
>
> ---

---

> ### Author Response · Authors · 2025-11-21
> **Response to Reviewer ZBoR (2/6)**
>
> > W-2 The construction of “rejected” responses using embedding similarity and the validation of human annotations with GPT-4 risk circular reasoning, as embedding distance and model agreement do not necessarily capture true cultural contrast or correctness.
> >
>
> Thank you for your kind suggestion. We address this by clarifying the existing validation and further presenting new empirical evidence involving native speakers to validate that our pipeline captures true cultural contrast.
>
> - **Existing Validation (Lines 171–178, Apdx. B.3):** Our original submission has already shown that **human and GPT-4o exhibit high agreement** on the cultural alignment of constructing “chosen” responses and the cultural inaccuracy of “rejected” responses. Additionally, it also demonstrate that **GPT-4’s judgments are strongly correlated with human annotations**. This initial evidence suggests that the model serves as a reliable proxy for human cultural evaluation rather than a source of circular logic.
> - **Native speaker validation**: To more directly address concerns regarding cultural contrast and the accuracy of our overall construction process (embedding similarity combined with GPT-4o judgments), we conducted a new native-speaker evaluation during the rebuttal period, using the same annotators referenced in our response to W-1. These annotators assessed cultural alignment for accepted completions and cultural inaccuracies for rejected ones across 1,000 randomly selected samples—100 samples per language, each containing 25 instances representing every cultural dimension.
>
>     **Results**: The results in the following table show that **native annotators achieved a high agreement ratio with our automated construction method**. The high agreement rates verify that **our constructions align closely with native-level cultural preference standards**, thereby directly addressing the concern.
>
>     | Language | Chosen Completion Agreement (%) | Rejected Completions Agreement (%) |
>     | --- | --- | --- |
>     | Spanish | 94.0 | 92.3 |
>     | German | 91.0 | 91.0 |
>     | Thai | 99.0 | 93.3 |
>     | Vietnamese | 99.0 | 99.3 |
>     | Korean | 87.0 | 84.0 |
>     | Chinese | 93.0 | 90.3 |
>     | Arabic | 84.0 | 77.0 |
>     | Russian | 84.0 | 86.7 |
>     | English | 95.0 | 93.7 |
>     | Japanese | 93.0 | 80.0 |
>
> **Action taken**: We have revised the manuscript to include this native-level annotation experiment in **Section 3** and details in **Appendix B.3** to highlight the strong agreement between our construction process and native-level annotators.
>
> ---
>
> > W-3 During generation of the dataset (Figure 14), it appears to assume a one-to-one correspondence between language and culture, which would overlook multilingual or multicultural contexts where a single language represents diverse cultural norms.
> >
>
> Thank you for your kind suggestion. We address this concern in the following two parts:
>
> - **Grounded in established literatures**: in We clarify that this approach is grounded in established methodology rather than simplistic one-to-one assumptions. As detailed in **Appendix A.1 (Lines 1258-1262)**, our methodology grounds language-to-culture mapping in established frameworks from cultural sociology [1][2][3]. **These studies validate the use of language as a primary, robust proxy for cultural norms in large-scale analysis.** Consequently, our approach **aligns with standard practices in cross-cultural NLP research [4][5][6]**, balancing the need for computational feasibility with theoretical validity.
> - **Empirical experiments validation**: Additionally, to directly and empirically address the concern regarding multicultural ambiguity, we conducted a robustness experiment comparing outputs from our original prompts (implicit cultural framing via language) against prompts with explicit country/culture specifications. Using Qwen-3-8B-Embedding, we measured cosine similarity across 10 languages spanning diverse cultural regions. The results demonstrate consistently high alignment (76.9%–92.7% similarity), **confirming that implicit linguistic framing effectively captures the intended cultural context without introducing substantial drift**.
>
>
>     | Language | Cosine Similarity (%) |
>     | --- | --- |
>     | Spanish | 84.7 |
>     | German | 85.1 |
>     | Thai | 79.3 |
>     | Vietnamese | 76.9 |
>     | Korean | 83.8 |
>     | Chinese | 92.7 |
>     | Arabic | 78.6 |
>     | Russian | 90.6 |
>     | English | 89.2 |
>     | Japanese | 91.2 |
>
> **Action taken**: We have revised the manuscript to include this robustness experiment in **Appendix B.1** to strengthen the reliability of the construction process.
>
> ---

---

> ### Author Response · Authors · 2025-11-21
> **Response to Reviewer ZBoR (3/6)**
>
> > W-4 Some cultural values and commonsense knowledge evolve over time, which potentially misrepresenting shifting cultural norms.
> >
>
> We explicitly account for the temporal evolution of cultural values by anchoring our benchmark in the most recent high-quality sources available: *Cultural Atlas* (~-2025) and *World Values Survey Wave 7* (2017–2022). Crucially, these datasets postdate the training cutoffs of many evaluated LLMs (typically pre-2023)[16][17]. This methodology provides a strict "temporal advantage," allowing us to evaluate whether models align with contemporary norms or regress to outdated patterns. By prioritizing fresh, human-curated data, our approach directly mitigates the risk of misrepresenting shifting values, consistent with cross-cultural NLP best practices [4, 7, 8].
>
> Finally, while we agree that cultural nuances shift over time, core cultural values and commonsense reasoning tend to exhibit relative stability over short-to-medium time horizons. We treat our current work as a foundational baseline. To mitigate the risk of temporal misalignment, we plan to maintain this line of research and update the benchmark to capture gradual sociological shifts.
>
> **Action Taken:** We have revised **Section 3** to explicitly emphasize that these data sources represent the most recent and authentic references for contemporary cultural values and commonsense knowledge.
>
> ---
>
> > W-5 Finally, the cultures being considered are limited to languages that are relative high/medium-resourced languages. However, I would say 10 cultures/languages are quite a lot already if the dataset is assessed by native people.
> >
>
> We appreciate the reviewer acknowledging the substantial challenge of rigorously assessing 10 cultures. As a pioneering "0-to-1" benchmark for assessing cultural awareness of reward models, we prioritized cultural evaluation data validity and the availability of qualified native annotators over broad but potentially unreliable culture coverage. Focusing on these representative high/medium-resource cultures allowed us to ensure strict quality control through native verification (as detailed in **Response to W-1 & W-2**). We believe this rigorous foundation is the first step before expanding to lower-resource regions in future work.
>
> ---

---

> ### Author Response · Authors · 2025-11-21
> **Response to Reviewer ZBoR (4/6)**
>
> > W-6 While I agree that m-reward-bench does not provide multicultural evaluation, I don't think correlation of performance between one benchmark and another is a good indicator of a well-correlated benchmark in terms of cultural knowledge, as there could be many confounding factors that represent such correlation. In addition, it only measures linear correlation, doesn't necessarily mean there is no correlation, furthermore in terms of causality that CARB represents multicultural benchmark.
> Suppose correlation were a good measure. Notice that include-base-44 itself consists of academic and professional exams, which also includes cultural-agnostic questions, AlpacaEval (which is the source of OMGEval) contain a lot of questions that are not culturally related (as shown in small number of localizations in the original paper), and BLEND's answers depend on the annotations being given (not representative of the culture of a country as a whole), so another reliance on GPT-4 as a judge wouldn't be great. If correlation were a good measure, this wouldn't remove the possibility that the performance could be correlated to the cultural-agnostic questions.
> >
>
> We appreciate the reviewer's rigorous scrutiny regarding potential confounders in correlation analysis. We respectfully clarify that while correlation ≠ causality, it is **the standard and practical validation method established in recent reward benchmark literature[9][10]**. While we do not claim strict causality, adhering to this methodology ensures our evaluation remains comparable to prior work and provides a practical, widely accepted signal of benchmark validity.
>
> Furthermore, we wish to address your specific concerns about confounders regarding **general capability** and **GPT** **judge bias** through two new control experiments:
>
> - **On cultural-agnostic questions:** We first clarify that **the correlations reported in the original paper were calculated on the Culturally Relevant subset**. To explicitly rule out "general capability" as a confounder, we extended the Best-of-N Sampling analysis (using Aya-Expanse-8B, consistent with **Sec. 5.1**) across three distinct partitions. Testing three subsets (full, cultural-relevant, cultural-agnostic) reveals CARB correlates strongly (0.83/0.75/0.78) *only* on the cultural-relevant subset, where M-Rewardbench performs poorly (0.31/0.33/0.37). The reversed pattern on cultural-agnostic questions confirms our correlation captures cultural knowledge, not general performance.
>
>
>     | include-base-44/BLEnD/OMGEval | Full set | Culturally Relevant Set (the original settings in the paper) | Culturally Agnostic Set |
>     | --- | --- | --- | --- |
>     | M-Rewardbench | 0.58/0.55/0.59 | 0.31/0.33/0.37 | 0.74/0.79/0.84 |
>     | CARB | 0.62/0.47/0.51 | 0.83/0.75/0.78 | 0.33/0.33/0.28 |
> - **On GPT-4 judge bias:** We extended the RLHF analysis (consistent with **Sec. 5.2**) to compare **GPT-4o** against **native human judges**. CARB maintains high correlation under human evaluation (0.59/0.62) compared to M-RewardBench (0.08/0.06). This consistency confirms that our findings are robust to the choice of judge and not an artifact of GPT-4o bias.
>
>
>     | OMGEval/BLEnD | GPT-4o Rating | Native-speaker Rating |
>     | --- | --- | --- |
>     | M-Rewardbench | 0.039/0.022 | 0.082/0.063 |
>     | CARB | 0.649/0.680 | 0.589/0.622 |
>
> These new experiments provide empirical evidence that **our correlation analysis is robust and a valid indicator of a well-correlated benchmark in terms of cultural knowledge, not driven by the potential confounders identified**.
>
> **Action taken**: We have revised the manuscript to include this additional experiment in **Section 5** and **Appendix D.6** to exclude the impact of potential confounders in the correlation analysis.
>
> ---

---

> ### Author Response · Authors · 2025-11-21
> **Response to Reviewer ZBoR (5/6)**
>
> > W-7 Section 5 claims strong “multilingual” correlation between CARB and downstream alignment, yet Section 6.2 explicitly shows that reward models exhibit cross-lingual inconsistency by assigning divergent scores to the same semantic content across languages. Wouldn't this show that CARB’s predictive validity about assessing non-shallow cultural features to be wrong?
> >
>
> Thank you for your kind suggestion. We clarify that there is no contradiction, as the two sections assess orthogonal properties of Reward Models (RMs). Specifically:
>
> - **Section 5** establishes a correlation based on **intra-language relative ranking** (Best-of-N accuracy). It measures the RM's ability to correctly distinguish the best response from rejected ones *within* a specific language context.
> - **Section 6.2**, conversely, identifies a failure mode in **absolute score consistency** across languages.
>
> Crucially, cross-lingual variance in absolute scores (Sec 6.2) does not negate the RM's ability to correctly rank candidates within a single language (Sec 5). Even if an RM exhibits a systematic bias (e.g., assigning lower absolute scores to all Chinese outputs compared to English), it can still successfully identify the superior response relative to inferior ones *within* that Chinese subset. Therefore, **the absolute scoring instability does not invalidate CARB’s predictive validity regarding cultural alignment**.
>
> **Action taken:** We will revise the manuscript to explicitly distinguish between these relative and absolute performance metrics.
>
> ---
>
> > W-8 While the result for the reward model framework looks promising on m-reward-bench and CARB, there is a lack of evidence that the reward model can be used to successfully post-train a policy model (RLHF or DPO).
> >
>
> We thank the reviewer for this valuable suggestion. To address this, we conducted Direct Preference Optimization (DPO) experiments using `Llama-3.1-Tulu-3-8B-SFT` as the base policy. We constructed preference datasets (derived from Alpagasus, Ultrafeedback [11][12], and CARE [4]) labeled by our `Think-as-Locals-7B` RM and compared this against a strong baseline using `Qwen2.5-7B-Instruct` as the RM.
> As shown in the table below, our reward model substantially outperforms the baseline across both **general capabilities** and **cultural alignment**. These results explicitly confirm that **the high performance of our proposed method on benchmarks effectively translates to superior downstream policy alignment**.
>
> | Model Variant (Reward Model used) | X-AlpacaEval (win rate against GPT-4-turbo evaluated by GPT-4) | MT-Bench (judged by GPT-4o) | BLEnD (judged by GPT-4o) | OMGEval (judged by GPT-4o) |
> | --- | --- | --- | --- | --- |
> | Base: Llama-3.1-Tulu-3-8B-SFT | 5.34% | 4.31 | 2.78 | 3.41 |
> | + DPO (Qwen2.5-7B-Instruct) | 14.6% | 6.02 | 4.22 | 5.87 |
> | + DPO (Think-as-Locals-7B) | **18.1%** | **6.54** | **5.18** | **6.46** |
>
> **Action taken**: We have revised the manuscript to include this additional LLM alignment experiment in **Section 7**. Thanks for your constructive consideration. This enhancement will better emphasize the contribution and significance of the proposed method.
>
> ---
>
> > Q-1 Related to **W1**, what kind of metric did you use to measure the inter-annotator agreement between GPT-4o since it's in percentage?
> >
>
> We calculate the **agreement ratio** between human annotators, and between human and GPT-4o on a randomly sampled subset. Additionally, we assessed GPT-4o's **self-consistency ratio** (majority vote stability) across the full sets. Both metrics demonstrated high consistency**,** confirming the reliability of our constructed CARB and GPT-as-judge.
>
> **Action taken**: We have revised **Table 12** to remove ambiguity.
>
> ---

---

> ### Author Response · Authors · 2025-11-21
> **Response to Reviewer ZBoR (6/6)**
>
> > Q-2 Most of the reward models used are LLM-as-a-judge(s) or reward model that is trained in English, how about specific multilingual models that (possibly) learn cultures during training (from recent related multilingual works), such as mR3 [1], Multilingual Nemotron [2], m-prometheus [3]?
> >
>
> We appreciate the reviewer’s suggestion to evaluate dedicated multilingual reward models. Following this recommendation, we conducted new experiments using mR3 [13], Multilingual Nemotron [14], and M-Prometheus [15].
>
> As detailed in the table below, our results reveal a critical distinction between *linguistic* proficiency and *cultural* awareness. While these models perform exceptionally well on general multilingual benchmarks (e.g., mR3-14B achieves **91.8** on M-RewardBench), their performance drops significantly on our cultural benchmark, CARB (e.g., mR3-14B scores only **59.8**).
>
> **This performance gap strongly validates our central thesis:** exposure to multilingual data does not inherently confer the ability to discern complex cultural nuances. These findings demonstrate that general multilingual models cannot replace the need for the explicit cultural alignment methods proposed in our work.
>
> | Models | M-RewardBench | CARB | Average |
> | --- | --- | --- | --- |
> | **Classifier-based RMs** |  |  |  |
> | Llama-3.3-Nemotron-70B-Reward-Multilingual | 87.4 | 63.9 | 75.7 |
> | **Generative RMs** |  |  |  |
> | M-Prometheus | 76.9 | 57.2 | 67.1 |
> | **Reasoning RMs** |  |  |  |
> | mR3-4B | 90.4 | 54.3 | 72.4 |
> | mR3-8B | 89.4 | 57.7 | 73.6 |
> | mR3-14B | 91.8 | 59.8 | 75.8 |
>
> **Action taken**: We have revised the manuscript to include this additional evaluation results in **Table 3**. Thanks for your constructive consideration. This enhancement will better emphasize the contribution and significance of the proposed method.
>
> ---
>
> > Q-3 After RLVF, seems like there is much more improvement in CARB compared to m-reward-bench, and I wonder if it's due to in-domain evaluation as the curated training dataset also involves CARB?
> >
>
> We confirm that the CARB dataset was strictly excluded from our training data. The performance discrepancy stems from the specific composition of our training mixture (general-purpose vs. culturally relevant data), rather than in-domain evaluation.
>
> To demonstrate this, we conducted an ablation study (see Table) comparing models trained on purely general data (Ultrafeedback, Alpagasus [11][12]) versus purely cultural data (CARE [4]). The results reveal distinct sensitivities: general data primarily boosts M-Rewardbench, whereas cultural data drives CARB performance. Consequently, our reported gains result from the synergistic effect of our mixed strategy, not data contamination.
>
> | Train data (Total 6k samples) | M-Rewardbench | CARB |
> | --- | --- | --- |
> | Qwen2.5-7B-Instruct | 77.1 | 62.6 |
> | multilingual Ultrafeedback & Alpagasus (2k samples each language) | 79.8 | 65.4 |
> | CARE (2k samples each language) | 77.9 | 67.9 |
> | Mix 1:1 | **80.1** | **69.2** |
>
> **Action taken**: We have revised the manuscript to include this controlled ablative study in **Appendix G.4**.
>
> ---
>
> **Reference**
>
> [1] Hofstede. Culture’s consequences. 2nd ed., 2001.
>
> [2] Teagarden et al. Culture, leadership, and organizations: The globe study. AMP, 2005. [
>
> 3] WVS. World values survey. 2022.
>
> [4] Guo et al. CARE: Multilingual Human Preference Learning for Cultural Awareness. EMNLP, 2025.
>
> [5] Myung et al. Blend: A benchmark for llms on everyday knowledge. NeurIPS, 2024.
>
> [6] Romanou et al. INCLUDE: Evaluating Multilingual Language Understanding with Regional Knowledge. ICLR, 2025.
>
> [7] Schneider et al. GIMMICK--Globally Inclusive Multimodal Multitask Cultural Knowledge Benchmarking. Findings of ACL, 2025.
>
> [8] Liu et al. OMGEval: An Open Multilingual Generative Evaluation Benchmark. arXiv:2402.13524, 2024.
>
> [9] Malik et al. RewardBench 2: Advancing Reward Model Evaluation. arXiv:2506.01937, 2025.
>
> [10] Zhou et al. RMB: Comprehensively benchmarking reward models in LLM alignment. ICLR, 2025.
>
> [11] Yang et al. Implicit Cross-Lingual Rewarding for Efficient Multilingual Preference Alignment. Findings of ACL, 2025.
>
> [12] Yang et al. Language Imbalance Driven Rewarding for Multilingual Self-improving. ICLR, 2025.
>
> [13] Anugraha et al. mR3: Multilingual Rubric-Agnostic Reward Reasoning Models. arXiv:2510.01146, 2025.
>
> [14] Wang et al. HelpSteer3-Preference: Open Human-Annotated Preference Data. arXiv:2505.11475, 2025.
>
> [15] Pombal et al. M-Prometheus: A Suite of Open Multilingual LLM Judges. arXiv:2504.04953, 2025.
>
> [16] Touvron et al. Llama 2: Open foundation and fine-tuned chat models. arXiv:2307.09288, 2023.
>
> [17] Qwen Team. Qwen2 technical report. arXiv:2407.10671, 2024.

---

### Official Review · Reviewer_wu8s · 2025-11-01

**Soundness:** 4
**Presentation:** 3
**Contribution:** 4
**Rating:** 6
**Confidence:** 4

**Summary:**

This paper introduces a high-quality dataset called the Cultural Awareness Reward Modeling Benchmark (CARB). By benchmarking a series of reward models on CARB, the authors identify spurious correlations in culture-aware reward modeling, which shows that RM scores often depend on surface-level features rather than genuine cultural understanding. To address this problem, they propose Think-as-Locals, a method designed to elicit deeper, culturally grounded reasoning in generative reward models through reinforcement learning with verifiable rewards. This approach helps mitigate the influence of spurious features and advances culture-aware reward modeling.

**Strengths:**

1. The paper introduces a high-quality benchmark, Cultural Awareness Reward Modeling Benchmark (CARB), which covers 10 distinct cultures and typologically diverse languages across four domains. This benchmark is a valuable contribution for evaluating the cultural awareness of reward models.
2. The authors conduct comprehensive experiments using a range of models and datasets, enabling a thorough comparison of different models’ capacities for cultural awareness.
3. The paper proposes Think-as-Locals, a method that improves reward model performance by over 10% while reducing correlations with surface-level features.

**Weaknesses:**

1. The paper attempts to cover too many aspects within a single work, without providing sufficient description or analysis for each section. For instance, Table 2 compares the performance of many models, but the accompanying explanation is brief and lacks analysis of why certain models outperform others.
2. The analysis in Section 4 appears redundant and somewhat unrelated to the main narrative of the paper; it could be moved to the Appendix.
3. Think-as-Locals is the core method proposed in this paper, yet the authors do not provide enough justification or explanation for its design choices.

**Questions:**

1. Line 159 – What are the reference embeddings referring to?
2. What is the definition of Classes 3, 4, and 5 in Figure 2?
3. Table 3 is somewhat confusing. Why does it include comparisons with many models that were not trained using the proposed method, instead of focusing only on models trained by the authors?

---

> ### Author Response · Authors · 2025-11-21
> **Response to Reviewer wu8s (1/2)**
>
> We sincerely thank you for providing thoughtful and constructive feedback. Please kindly find our response to your comments below. We have added an explicit and **systematic research trajectory** **in** **the Introduction and Section 4**, provided a **detailed analysis of Table 2** **in** **Section 4.2**, and offered a more **comprehensive explanation of the proposed method** and experimental setups in **Section 7**. All revisions made to the paper are highlighted in blue for your ease of reference. We hope that our response satisfactorily addresses the issues you raised. Please feel free to let us know if you have any additional concerns or questions :-)
>
> ---
>
> > W-1 The paper attempts to cover too many aspects within a single work, without providing sufficient description or analysis for each section. For instance, Table 2 compares the performance of many models, but the accompanying explanation is brief and lacks analysis of why certain models outperform others.
> >
>
> Thank you for the kind comment. We would like to address your concern regarding the following three parts:
>
> - **Overall coherent research trajectory**: Indeed, our work follows a single, coherent research trajectory: **(i) evaluation (Section 4) → (ii) diagnostic analysis (Sections 5, 6) → (iii) targeted improvement (Section 7)** of cultural-aware reward models. Each section builds directly on the previous one, and this progressive structure has also been noted by other reviewers as a strength (e.g., Reviewer SHu3’s comment: “*The paper is well written and easy to follow, with a clear logical flow: each section identifies a limitation and proposes a corresponding solution.*”).
> - **Regarding the brief explanation of Table 2**: initially, because of page limitations, Section 4 (“Evaluation”) is kept intentionally brief by moving additional experimental results explanation to **Appendix C.5**, allowing more main text space for the analyses in Sections 5 and 6 and for the methodological details presented in Section 7.
> - **To directly address concerns related to Table 2**, we have now expanded Section 4.2 with a focused discussion explaining *why* certain models outperform others. In particular, we highlight that generative RMs dominate the top rankings due to (1) their large-scale, broad pre-training—which provides the cultural and world knowledge required for nuanced evaluation—and (2) their stronger reasoning abilities compared to classifier-based RMs. We also analyze cross-lingual consistency patterns, noting that top generative RMs (e.g., Qwen3-235B-A22B-Instruct-2507) perform reliably across all cultures, whereas classifier RMs often excel only in high-resource languages.
>
> **Action taken**: We have revised the manuscript to strengthen the systematic research roadmap in the **Introduction section** and include this detailed explanation and discussion in **Section 4.2**. Thanks for your constructive comment. This enhancement will better emphasize the contribution and findings of CARB.
>
> ---
>
> > W-2 The analysis in Section 4 appears redundant and somewhat unrelated to the main narrative of the paper; it could be moved to the Appendix.
> >
>
> We would like to clarify that **Section 4 forms the essential** **“Evaluate” stage** of our structured *Evaluate → Diagnose → Improve* research framework (as detailed in response to W-1). It establishes the empirical baseline that all subsequent diagnostic analyses and improvements directly build upon. Retaining this in the main text **preserves the logical progression** and **clearly establishes the motivation** for subsequent sections.
>
> **Action taken**: To make this progression even more straightforward, we will strengthen **Section 4’s introduction** and the introductory roadmap in **Section 1**.
>
> ---

---

> ### Author Response · Authors · 2025-11-21
> **Response to Reviewer wu8s (2/2)**
>
> > W-3 Think-as-Locals is the core method proposed in this paper, yet the authors do not provide enough justification or explanation for its design choices.
> >
>
> Thank you for raising this point. We have expanded Section 7 to provide more details related to the proposed method, as summarized in the following:
>
> **Our method is directly motivated by an identified failure mode:** **Section 6.1 (Fig. 6)** shows that existing reward models rely heavily on ***spurious correlations***—surface cues such as cultural labels or phrasing—rather than the underlying cultural concepts. *Think-as-Locals* is designed specifically to correct this pathology through the following two design choices.
>
> 1. **We require the model to generate structured cultural-evaluation criteria *before* giving a judgment.** This shifts the model away from shortcut features and toward causal reasoning. Figure 8 empirically confirms that this design substantially increases sensitivity to causal perturbations while reducing sensitivity to spurious ones.
> 2. We adapt RLVR with a two-part reward: (1) **a correctness reward to ensure accurate final judgments**, and (2) **a criteria-quality reward to optimize the relevance and rigor of the intermediate reasoning**. Figure 9 shows that removing either component degrades performance, validating both design choices.
>
> **Action taken**: We have revised the manuscript to include a more detailed justification and explanation for our design choices of the proposed method in **Section 7**.
>
> ---
>
> > Q-1 Line 159 – What are the reference embeddings referring to?
> >
>
> The reference embeddings are the vector representations of the culturally relevant references mentioned in lines 156–158 (“Chosen completions were generated by providing top-tier LLMs with prompts and their corresponding culturally relevant references, which were collected concurrently with prompt sourcing.”). We encoded these references using the same embedding（qwen3 embedding） model to compute cosine similarity with the generated completion, filtering out low-quality responses below our similarity threshold. This process is clearly described in Section 3.2.
>
> **Action taken**: To avoid ambiguity, we have revised **Line 159** to explicitly replace the “reference” with the associated culturally relevant reference.
>
> ---
>
> > Q-2 What is the definition of Classes 3, 4, and 5 in Figure 2?
> >
>
> As stated in the Figure 2 caption, the Class 3, 4, and 5 categories are directly adopted from the established resource availability taxonomy of Joshi et al. (2020)[1]: Class 5 ("Winners"), Class 4 ("Underdogs"), and Class 3 ("Rising Stars"). A larger class number suggests greater availability of language data. The complete language-to-class mapping is provided in **Table 4** (**Appendix A.1**) for complete transparency.
>
> **Action taken**: We have revised the caption of **Figure 2** to directly introduce the definition of Classes 3, 4, and 5.
>
> ---
>
> > Q-3 Table 3 is somewhat confusing. Why does it include comparisons with many models that were not trained using the proposed method, instead of focusing only on models trained by the authors?
> >
>
> This comparison is designed to serve a two-fold purpose.
>
> 1. The primary goal is to **directly isolate the contribution of our training method**. To this end, Table 3 provides direct comparisons by benchmarking a base model *before* and *after* our method is applied.
>     - For example, the baseline `Qwen2.5-7B-Instruct` model scores **62.6** on CARB.
>     - The same model trained with our `Think-as-Locals` method, listed as `Ours (Based on Qwen2.5-7B-Instruct)`, scores **78.8**.
>
>     This direct comparison, which holds for the 14B and 32B variants as well, demonstrates the significant and consistent performance gains attributable specifically to our approach.
>
> 2. The second purpose of the table is to **situate our method within the broader SOTA landscape**. To do this, we compare our results against established, representative models from all three major reward modeling paradigms (Classifier, Generative, and Reasoning). We used the officially released checkpoints for some of these baselines because their training code is often not open-source.
>
> In summary, the table is structured to show both (1) that our method *works* (via direct, controlled comparisons) and (2) that the results *matter* (by outperforming current SOTA models).
>
> **Action taken**: We have revised the manuscript to include a detailed explanation of this comparison setup **in** **Section 7**.
>
> ---
>
> **Reference**
>
> [1] Pratik Joshi, Sebastin Santy, Amar Budhiraja, Kalika Bali, and Monojit Choudhury. 2020. [The State and Fate of Linguistic Diversity and Inclusion in the NLP World](https://aclanthology.org/2020.acl-main.560/). In *Proceedings of the 58th Annual Meeting of the Association for Computational Linguistics*, pages 6282–6293, Online. Association for Computational Linguistics.

---

### Author Response · Authors · 2025-11-24
**A Friendly Reminder: A Polite Follow-Up Regarding Our Rebuttal**

Dear Reviewers and AC,

I hope this message finds you well. Thank you sincerely for your time and consideration.

Due to an unexpected issue, the rebuttal we submitted before the AOE deadline on November 20 may not have been delivered by email. We are posting this comment to ensure smooth and transparent communication. We sincerely hope that our responses address your concerns. If any questions remain or if further clarification would be helpful, we would be very glad to continue the discussion.

Thank you once again for your time and valuable feedback throughout this process.

Best regards,

All Authors

---

### Meta-Review · Area_Chair_Lw8v · 2026-01-08

**Summary:**

This paper addresses the critical gap in evaluating and improving the cultural alignment of Reward Models (RMs), which serve as the proxy for human preference in RLHF. The reviewers initially raised several valid concerns, which were largely mitigated during the rebuttal phase, including the Reliance on Synthetic Data, Reviewer Qualifications & Judge Reliability, Methodological Details and correlations between CARB scores and downstream policy performance.

**Reviewer Concerns:**

1) Reliance on Synthetic Data (Reviewers ZBoR, SHu3, 9wq4): A primary concern was the heavy reliance on GPT-4o for dataset generation and translation, raising fears of "translationese" or circular reasoning where models evaluate their own biases.

Resolution: The authors responded by incorporating authentic human-generated examples and conducting native-speaker verification (using undergraduate/graduate students from the respective cultural backgrounds). This significantly strengthened the validity of CARB. Reviewers 9wq4 and SHu3 explicitly raised their scores (to 8) based on these additions.

2) Concerns were raised regarding whether the human annotators were native speakers and whether LLM-as-a-Judge is reliable for cultural nuances.

Resolution: The authors confirmed the native-speaker status of annotators and validated the judge's reliability against human baselines, satisfying the reviewer.

3) Methodological Details & Justification (Reviewer wu8s): Questions regarding the analytical depth and the design choices of ``Think-as-Locals.''

Resolution: The authors provided detailed model analyses and clarified the design choices in the revision. While Reviewer wu8s did not provide a post-rebuttal update, the clarifications align with the positive reception from other reviewers regarding the method's effectiveness.

**Reviewer Scores:**

Following our Nov 20 (AOE) response, the ratings of the submission improved to a snapshot of 8, 8, 6, 4 (as of Nov 26, especially before the platform issue).

---

### Decision · Program_Chairs · 2026-01-26

Accept (Poster)